# Willingness to participate in future HIV vaccine trials among men who have sex with men and female sex workers living in Nairobi, Kenya

**Elizabeth Mueni Mutisya**[1]*, **Gaudensia Mutua**[1], **Delvin Nyasani**[1], **Hannah Nduta**[1], **Rhoda W. Kabuti**[2], **Vincent Muturi-Kioi**[3], **Gloria Omosa-Manyonyi**[4], **Andrew Abaasa**[5], **Krysia Lindan**[6], **Matt A. Price**[6,7], **Joshua Kimani**[2], **Aggrey Omu Anzala**[1]

**1** KAVI-Institute of Clinical Research, University of Nairobi, Nairobi, Kenya, **2** SWOP-Kenya, University of Nairobi/University of Manitoba, Nairobi, Kenya, **3** IAVI, Nairobi, Kenya, **4** School of Medicine, College of Health Sciences, University of Nairobi, Nairobi, Kenya, **5** MRC/UVRI and LSHTM Uganda Research Unit, Entebbe, Uganda, **6** Department of Epidemiology and Biostatistics, University of California, San Francisco, California, United States of America, **7** IAVI, New York, New York, United States of America

* Emutisya@kaviuon.org

**Data Availability Statement:** All data underlying the study can be downloaded from URL https://doi.org/10.3886/E119622V1.

## Abstract

### Objective

To evaluate factors associated with willingness to participate in future HIV vaccine trials among men who have sex with men and female sex workers living in Nairobi, Kenya.

### Background

Working with 'key populations', those at elevated risk of HIV acquisition, is important to conduct efficient HIV prevention trials. In Nairobi Kenya, HIV infection is higher in men who have sex with men (MSM) and female sex workers (FSW) than in the general adult population, hence the need to establish if they would be willing to participate in future HIV vaccine trials.

### Methods

We administered a structured questionnaire to MSM and FSW enrolled in a simulated vaccine efficacy trial (SiVET). The SiVET was an observational study designed to mimic the rigors of a clinical trial to assess HIV risk characteristics at baseline. After 12–15 months of follow-up, a structured questionnaire was administered to evaluate hypothetical willingness to participate in future HIV vaccine trials.

### Results

Of 250 persons (80% MSM by design) enrolled in SiVET, 214 attended the final study visit and 174 (81%) of them expressed hypothetical willingness to participate in future HIV vaccine trials. These were 82% of MSM and 80% of FSW of those who attended the final study visit. Having a very good experience in the SiVET trial predicted willingness to participate in

**Funding:** This work was funded by IAVI, with generous support of USAID and other donors; a full list of IAVI donors is available at www.iavi.org. The contents of this manuscript are the responsibility of KAVI-ICR and co-authors and do not necessarily reflect the views of USAID or the US Government. The manuscript was developed through the support from the University of California, San Francisco's International Traineeships in AIDS Prevention Studies (ITAPS), U.S. NIMH, R25MH064712.

**Competing interests:** The Authors declare that they have no competing interests.

future HIV vaccine trials. Motivating factors for participation included a desire to receive education about HIV (59%) and to receive healthcare (57%).

## Conclusions

Our data demonstrate high willingness among key populations in Kenya, to participate in future HIV vaccine trials after completing participation in a SiVET. The findings suggest that these groups might be a reliable target population for consideration in future HIV vaccine trials. Assessment of willingness to participate in these populations provides important information that may help to inform future education and recruitment efforts for vaccine trials. Improving the research experience for members of key populations could impact their willingness to participate in HIV vaccine trials.

## Introduction

The world continues to experience the challenge of the HIV/AIDS pandemic; based on 2018 estimates, 37.9 million people are living with HIV around the world with 67% of these people living in sub-Saharan Africa and 1.6 million people in Kenya [1–5]. Interventions for prevention and treatment of HIV, including behavioural, structural and biomedical options, have become increasingly available and helped to combat the pandemic [1, 2, 5].

Over the past three decades there has been a drop in the annual number of new HIV infections, from a peak of 2.9 million globally in 1997 to 1.7 million in 2018 and a decline in the global incidence–prevalence ratio, from 11.2% in 2000 to 4.6% in 2018. Despite the overall decline in new HIV infections, sub-Saharan Africa continues to contribute the majority of incident cases with 800,000 new infections in 2018 [3, 5, 6]. Progress is slower than what is required to reach the world 2020 milestone of less than 500,000 new infections [1, 2, 5].More than half of the new infections are among members of key populations, and their sexual partners, who are at a higher risk of HIV infection compared to the general population. Men who have sex with men (MSM) and sex workers are 21times more likely to acquire HIV infection compared to adults in the general population and account for 17% and 6% of the new global infections [5]. In Kenya, among MSM, the incidence has been reported at 10.9 per 100 person-years of observation [7] and among female sex workers (FSW), at 5.6 per 100 person-years of observation [8]. This is estimated to account for 14–15% of all new infections in the country [9, 10]. The high incidence results in a relatively high national HIV prevalence, among FSW at 29.3% and MSM at 18.2% [10].

The high HIV incidence among key populations is indicative of an unmet medical need for HIV prevention services including the need for a safe and effective HIV vaccine to end the pandemic [4, 11]. Prioritizing key populations for prevention and treatment is likely to have a significant effect on reducing HIV transmission and the national burden of disease in Kenya [6, 12]. Key populations are suitable for inclusion in efficacy trials as the primary beneficiaries of an effective HIV vaccine [11, 13].

To achieve the objectives of any HIV prevention clinical trial, understanding logistic and statistical power considerations and ensuring participants are motivated to complete the study are all key issues [14]. Preparedness studies can be important precursors but 'willingness to participate' studies do not always predict actual participation, perhaps in part because it is hard to convey the commitment necessary to the potential participants [15, 16]. Simulated

Vaccine Efficacy Trials (SiVETs) are observational studies that have been designed to mirror the procedures of actual HIV vaccine trials. SiVETs enable researchers to gauge important elements in clinical trials including; recruitment, enrolment, retention of the suitable population, participant compliance with protocol procedures, willingness to participate in the trials and evaluate potential barriers to participation [17, 18]. SiVETs have been performed among high risk populations in several countries including Uganda [13, 19], Zambia [20] and South Africa [18]. Such trials have not previously been conducted among MSM and FSW in Kenya.

Between Oct 2017 and Sep 2018, we interviewed MSM and FSW who had completed participation in a SiVET in Nairobi, Kenya, to determine their willingness to participate in future HIV vaccine trials.

## Methods

### Setting

Potential participants were recruited from the Sex Workers Outreach Program (SWOP-Kenya) clinics serving FSW and MSM in Nairobi after community engagement and demand creation activities. The program offers services including HIV risk reduction counselling, anti-retroviral treatment (ART), HIV pre-exposure prophylaxis (PrEP), condoms and family planning services and diagnosis and treatment of sexually transmitted infections (STI) in an FSW and MSM friendly setting. Participants were referred and escorted from SWOP-Kenya clinics to the KAVI-Institute of Clinical research (KAVI-ICR) of the University of Nairobi.

### Sample size

A sample size of 250 for the SiVET was calculated based on being able to estimate a one-year retention rate of 80% with a precision of ±5%. The target was to enrol 200 MSM and 50 FSW. The choice to enrol 20% FSW was deliberate because HIV incidence is higher among Kenyan MSM than Kenyan FSW; we also wished to avoid stigma and any hard feelings among our partners at SWOP-Kenya and agreed that enrolling some women would be appropriate.

### Recruitment and screening

SWOP-Kenya clinic staff and affiliated peer outreach workers recruited potential participants from the clinics and hot spots, providing background information about the SiVET study. Hot spots are areas such as streets, bars, hotels or massage parlours in Nairobi where sex workers operate. HIV-uninfected persons were asked to participate in a study that entailed the delivery of hepatitis B vaccination to mirror the requirements of a HIV vaccine efficacy trial requiring follow up for 12 to 15 months (follow up was initially set to 15 months, but later amended to 12 months due to logistical and funding reasons). Those interested were pre-screened and then referred to KAVI-ICR for more information about the trial, an assessment of understanding of the study requirements, after which participants signed informed consent forms.

KAVI-ICR staff screened participants on the following eligibility criteria: age 18 years and above, residing in Nairobi, sexually active in the preceding 3months, willing to undergo HIV risk assessment and HIV testing, able to provide contact information, willing to be contacted by study staff and willing to return for study visits. FSW had to be actively involved in sex work for at least 3 months before screening and agree to use long-acting reversible non-barrier contraceptives such as injectable and implantable hormonal contraceptives, intrauterine devices or had a documented tubal ligation. We excluded participants who were HIV infected, had active hepatitis B infection, were known to be pregnant or were nursing mothers, had prior severe reactions to vaccines or had any significant clinical condition as assessed by the investigator.

## Study data

At enrolment, participants completed an interviewer-administered questionnaire on sociode-mographics, HIV risk behaviour and knowledge of HIV prevention methods. Information on their past medical history, including vaccinations and adverse reactions to vaccines, was also obtained.

Sociodemographic data included age and the highest level of education achieved. For the assessment of HIV risk behaviour we asked about: Frequency of alcohol consumption, the number of sex partners and new sex partners, use of any illicit drugs, practice of insertive or receptive anal sex and frequency of condom use in the preceding month. We administered an alcohol use disorder screening test, the CAGE [21] instrument, to classify participants.

We evaluated HIV prevention knowledge by asking participants: '*How do you think HIV/ AIDS can be prevented from an infected person to an uninfected person?*' Participants' responses were then scored using a predetermined list of correct ways to prevent HIV infection catego-rized into behavioural and biomedical prevention methods. Biomedical prevention methods included pre-exposure prophylaxis, post-exposure prophylaxis, prevention of mother to child transmission, treatment as prevention, and voluntary medical male circumcision. Behavioural prevention methods included abstinence, condom use, monogamy, limiting the number of sexual partners, use of clean needles by injecting drug users, and avoiding sharing razors/ blades/injection needles.

At the participants' last visit, those who were HIV negative completed an interview ques-tionnaire on willingness to participate in HIV vaccine trials. While those who were HIV infected were excluded from the interview. We asked the following question: '*Would you be willing to participate in a future HIV vaccine trial if it had exactly the same procedures as the SiVET study?*' Possible responses were '*yes very likely*', '*maybe yes*', '*maybe not*', '*no not at all*', and '*don't know*'. Participants gave reasons for their responses.

## Statistical methods

Data were captured and managed in OpenClinica version 3.0, transferred and analysed in STATA version 14.0. Overall and stratified (MSM and FSW) data on sociodemographics, HIV risk behaviour, HIV prevention knowledge and their SiVET experience were summarised using frequency and percentages. Participants were classified as willing to participate in future HIV vaccine trials if their response was 'yes very likely' or 'maybe yes' to participate in an HIV vaccine efficacy trial designed in the same way as the SiVET. We used chi-square tests to assess the relationship between participants' willingness to participate and their sociodemographic characteristics, HIV risk behaviour, HIV prevention knowledge and their SiVET experience for both the overall data and the data stratified by population (MSM and FSW).

Univariate logistic regression was used to evaluate the association of all variables with will-ingness to participate; factors associated with willingness to participate at p<0.20 on log likeli-hood test were included in the initial multivariable model (except for age and trial population that were included a priori). Factors were retained in the multivariable model if the log likeli-hood test p-value of inclusion of a factor was less than 0.05.

## Ethical statement

The study protocol was approved by the research ethics committee of Kenyatta National Hos-pital and University of Nairobi (KNH-UoN ERC; Ethical approval number P137/03/2015). Written informed consent to participate in the SiVETstudy was provided by all the partici-pants before screening. All participants with confirmed HIV infection were referred to their primary SWOP-Kenya clinic or other local HIV care and treatment facilities if preferred.

Participants were reimbursed 800 Kenya shillings (approximately US $8) for their time and effort at each scheduled study visit.

## Results

### Recruitment, screening and enrolment

A total of 739 potential participants were referred from SWOP-Kenya clinics to KAVI-ICR (Fig 1). Of these, 368 (49.8%) were eligible for screening for the SiVET and 250 (68%) met the eligibility criteria and were enrolled. The main reasons for exclusion at screening were: active hepatitis B infection [HBsAg+ (n = 14)], pre-existing medical conditions (n = 41) and potential participants declining enrolment (n = 47). After 12–15 months of study follow-up 87% remained on study; 217 (87%) completed the SiVET while 33(13%) had discontinued. The primary reasons for not completing study follow-up included: HIV infection, loss to follow-up and relocation outside of Nairobi (the study area).

Therefore, 219 (217 completed SiVET and 2 who had their final study visit before end of trial) were eligible for the willingness to participate in a future HIV vaccine trial interview.

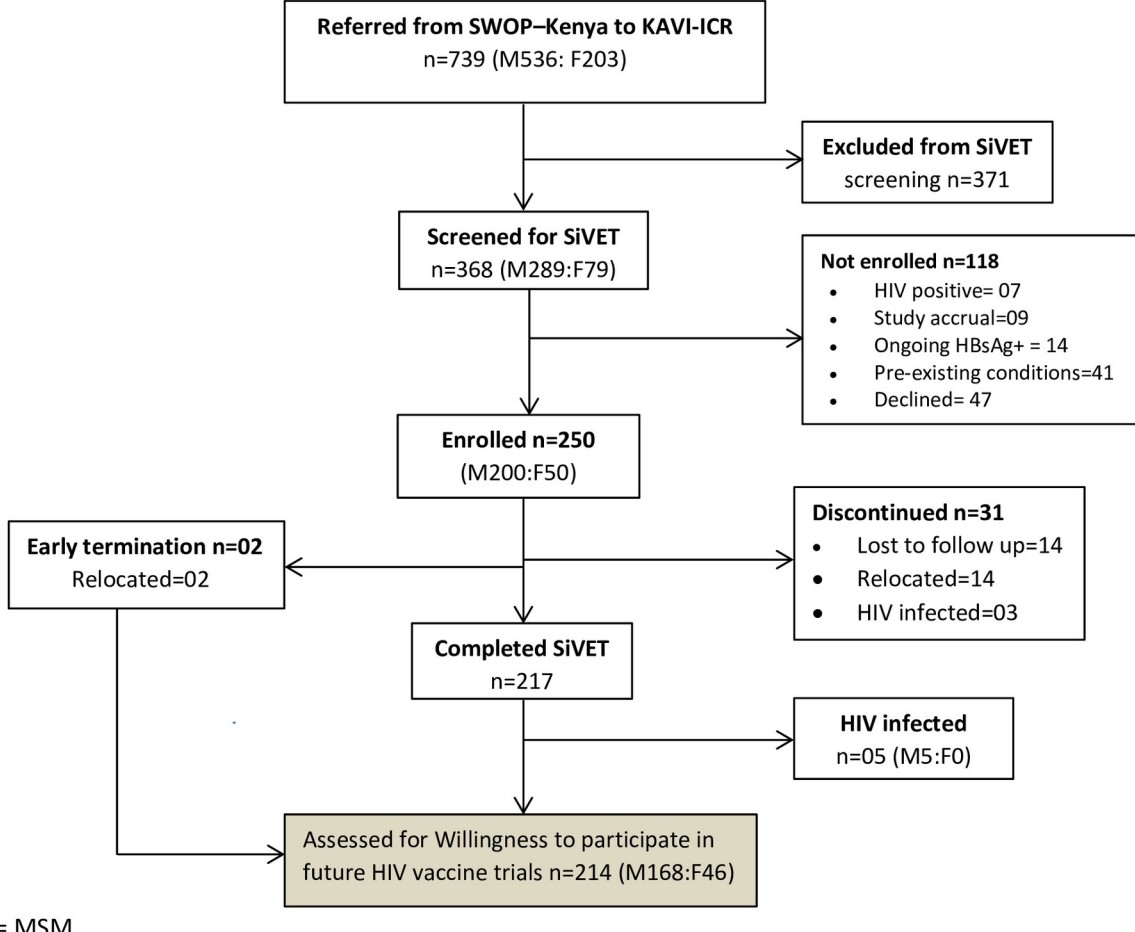

M= MSM

F=FSW

**Fig 1. Flow diagram of participant recruitment, screening, enrollment into SiVET study and assessment of willingness to participate in future HIV vaccine trials.** M = MSM, F = FSW.

Overall, 214(86%) participants completed the interview and were included in the analysis, while 5 were excluded from the analysis because of HIV infection.

## Participant characteristics and willingness to participate in future HIV vaccine trials

At their final study visit, 174 (81%) (Table 1) participants expressed willingness to participate in preventive HIV vaccine trials in the future. The proportion was similar among MSM 137 (82%) and FSW 37(80%). Fourty (19%) participants were not willing to participate and the main reason given was time commitment, fear that their HIV status would be disclosed to others without their permission and the strict contraception requirements.

## Predictors of willingness to participate in future HIV vaccine trials

Table 2 shows correlates of willingness to participate in an HIV vaccine trial, individually in bivariate analyses and in a final multivariable model. In the adjusted analysis, reporting a very good experience (compared to a good experience, as none of the participants reported a poor experience) in the SiVET study (aOR 5.54, 95% CI 2.39, 12.89, P-value <0.001) and reporting always using protection with condoms during sexual intercourse (aOR 2.53, 95% CI 0.99, 6.45 p-value 0.050) were associated with increased willingness to participate in a future HIV vaccine trial. Whereas knowledge of one to two or three to four HIV/AIDS biomedical prevention methods were less likely to be willing to participate compared with those with no knowledge (aOR 0.38, 95% CI 0.15, 0.93 P-value 0.035 and aOR 0.30, 95% CI 0.09, 0.93 P-value 0.036, respectively).

## Factors that would motivate potential participation in future HIV vaccine trial

More than half of the participants reporting willingness to participate in future HIV vaccine trials reported getting education about HIV (59%) or getting access to healthcare services (57%) as factors that motivated their participation (Table 3); other motivating factors reported included altruism, access to HIV counselling and testing services, hope of being protected against HIV, relationships with the clinic staff, and the need to stay busy.

## Discussion

Among MSM and FSW from Nairobi who had completed a SiVET trial, about four of five said they would be willing to participate in a future HIV vaccine trial. These findings suggest that these groups might be a reliable target population to consider in future HIV vaccine efficacy trials.

Our study found three factors significantly associated with willingness to participate in future HIV vaccine trials. Overall, most of the participants reported that they had a 'very good' experience in the SIVET, and no one reported a bad experience. Those whose experience was better (i.e., 'very good' vs. 'good') were more likely to express willingness to participate in future vaccine trials. Exploring factors that could improve participant experiences, and the potential barriers and motivators to participation could guide best practice and address concerns expressed by members of key populations engaged for participation in HIV vaccine trials. Participants only reported very good and good experiences; because feedback was collected by the same staff that had interacted with the participants over the period of the study, this is a possible source of response bias.

**Table 1. Characteristics of 214 participants enrolled in a simulated HIV vaccine trial in Nairobi and the proportion willing to enroll in a future HIV vaccine trial, Nairobi, Kenya; Sep 2015-Sep 2018.**

| Characteristic | All | | Men who have sex with men (MSM) N = 168 | | | | Female sex workers (FSW) N = 46 | | | |
|---|---|---|---|---|---|---|---|---|---|---|
| | (N = 214) | | (N = 168) | | Willing to Participate | | (N = 46) | | Willing to participate | |
| | N | (%) | N | (%) | N | (%) | N | (%) | N | (%) |
| Total | 214 | (100) | 168 | (79) | 137 | (82) | 46 | (21) | 37 | (80) |
| **Age, years** | | | | | | | | | | |
| 18–24 | 89 | (42) | 81 | (48) | 64 | (79) | 08 | (17) | 08 | (100) |
| 25–34 | 72 | (34) | 54 | (32) | 44 | (81) | 18 | (39) | 16 | (89) |
| ≥35 | 53 | (24) | 33 | (20) | 29 | (88) | 20 | (44) | 13 | (65) |
| **Highest level of education Completed** | | | | | | | | | | |
| Primary | 40 | (19) | 19 | (11) | 15 | (79) | 21 | (46) | 16 | (76) |
| Secondary | 116 | (54) | 95 | (57) | 78 | (82) | 21 | (46) | 19 | (90) |
| Tertiary/Higher | 58 | (27) | 54 | (32) | 44 | (81) | 04 | (08) | 02 | (50) |
| **Alcohol use in the last month** | | | | | | | | | | |
| Never | 77 | (36) | 63 | (37) | 53 | (84) | 14 | (30) | 14 | (100) |
| Sometimes | 137 | (64) | 105 | (63) | 84 | (80) | 32 | (70) | 23 | (72) |
| **Alcohol before sex in the last month** | | | | | | | | | | |
| Never | 133 | (62) | 110 | (65) | 92 | (84) | 23 | (50) | 22 | (96) |
| Sometimes | 81 | (38) | 58 | (35) | 45 | (78) | 23 | (50) | 15 | (65) |
| **Alcohol CAGE score[†]** | | | | | | | | | | |
| ≤1 | 189 | (88) | 149 | (89) | 123 | (83) | 40 | (87) | 33 | (83) |
| ≥2 | 25 | (12) | 19 | (11) | 14 | (74) | 06 | (13) | 04 | (67) |
| **Illicit drug use in the last month** | | | | | | | | | | |
| Yes | 160 | (75) | 118 | (70) | 94 | (80) | 42 | (91) | 35 | (83) |
| No | 54 | (25) | 50 | (30) | 43 | (86) | 04 | (09) | 02 | (50) |
| **Number of sex partners in the last month** | | | | | | | | | | |
| ≤1 | 59 | (28) | 52 | (31) | 45 | (87) | 07 | (15) | 05 | (71) |
| 2 | 47 | (22) | 41 | (24) | 32 | (78) | 06 | (13) | 05 | (83) |
| ≥3 | 108 | (50) | 75 | (45) | 60 | (80) | 33 | (72) | 27 | (82) |
| **New sex partners in the last month** | | | | | | | | | | |
| 0 | 106 | (50) | 97 | (58) | 80 | (82) | 09 | (20) | 08 | (89) |
| 1 | 34 | (16) | 27 | (16) | 21 | (78) | 07 | (15) | 07 | (100) |
| ≥2 | 74 | (34) | 44 | (26) | 36 | (82) | 30 | (65) | 22 | (73) |
| **Insertive Anal sex in the last month** | | | | | | | | | | |
| No | 39 | (23) | 39 | (23) | 35 | (90) | n/a | n/a | n/a | n/a |
| Yes | 129 | (77) | 129 | (77) | 102 | (79) | n/a | n/a | n/a | n/a |
| **Receptive anal sex in the last month** | | | | | | | | | | |
| No | 137 | (64) | 91 | (54) | 70 | (77) | 46 | (100) | 37 | (80) |
| Yes | 77 | (36) | 77 | (46) | 67 | (87) | 00 | (00) | 00 | (00) |
| **Condom use in the last month** | | | | | | | | | | |
| Sometimes | 43 | (20) | 40 | (24) | 31 | (78) | 03 | (07) | 01 | (33) |
| Always | 171 | (80) | 128 | (76) | 80 | (80) | 43 | (93) | 27 | (87) |
| **Number of known HIV behavioral Preventive methods** | | | | | | | | | | |
| **0** | **00** | **(00)** | **00** | **(00)** | **00** | **(00)** | 00 | (00) | 00 | (00) |
| 1 | 35 | (16) | 30 | (18) | 25 | (83) | 05 | (11) | 05 | (100) |
| 2–3 | 116 | (54) | 87 | (52) | 45 | (85) | 29 | (63) | 11 | (79) |
| 4–6 | 63 | (30) | 51 | (30) | 41 | (80) | 12 | (26) | 08 | (67) |

(*Continued*)

**Table 1.** (Continued)

| Characteristic | All | | Men who have sex with men (MSM) | | | | Female sex workers (FSW) | | | |
| --- | --- | --- | --- | --- | --- | --- | --- | --- | --- | --- |
| | | | N = 168 | | | | N = 46 | | | |
| | (N = 214) | | (N = 168) | | Willing to Participate | | (N = 46) | | Willing to participate | |
| | N | (%) | N | (%) | N | (%) | N | (%) | N | (%) |
| **Number of known HIV biomedical preventive methods** | | | | | | | | | | |
| 0 | 81 | (38) | 71 | (42) | 62 | (87) | 10 | (22) | 09 | (90) |
| 1–2 | 92 | (43) | 72 | (43) | 28 | (78) | 20 | (43) | 12 | (92) |
| 3–4 | 41 | (19) | 25 | (15) | 13 | (77) | 16 | (35) | 10 | (77) |
| **Experience in SiVET Study** | | | | | | | | | | |
| Good | 93 | (43) | 74 | (44) | 51 | (69) | 19 | (41) | 14 | (74) |
| Very good | 121 | (57) | 94 | (56) | 86 | (91) | 27 | (59) | 23 | (85) |

[†] CAGE score of ≤1 was an indication of no alcohol problem and a score of ≥2 indicated of alcohol problem.

n /a = not applicable.

Those who reported more consistent condom use were more willing to participate in future HIV vaccine trials. It is possible that these participants were more aware of their HIV risk, and interested in reducing it through participation in a trial. However, we did not find any association between alcohol use, drug use or number of sex partners and willingness to participate.

We found that participants with knowledge of biomedical HIV prevention methods were less likely to be willing to participate in future HIV vaccine trials compared to those with no knowledge of available methods. Surprisingly, a significant proportion of participants had no knowledge of biomedical prevention methods against HIV even with the presence of campaigns promoting these methods in Kenya and at the SWOP clinics where participants were recruited from. This observation may be due to disconnect between the open-ended approach used in this study (i.e., our interviewers asked open ended questions to participants, without specific prompts) and the specific prevention methods provided. It would be useful to further explore and understand reasons for this observation and institute mitigating measures if required.

Several factors were reported as motivators for participation in future HIV vaccine trials. Access to health care services and information, also previously cited in similar studies in East Africa [19, 22], were reported as important motivating factors. Access to healthcare is a challenge in these communities [23]; clinical researchers have a responsibility to provide the highest available standard of HIV prevention services to populations at risk of HIV participating in HIV vaccine trials. It is important to continually highlight this unmet medical need in order to influence policy and increase access to essential services to underserved communities. Altruism was reported as a motivating factor by several participants in the SiVET, this lines up with a systemic literature review [16] and a study among MSM and high risk women in America [24] that found altruism as the major motivator to participate in HIV vaccine trials.

Among those who were not willing to participate, fear that their HIV status would be disclosed to others, or unwillingness to comply with the requirements for use of contraception during the trial were the main reasons given against future participation. In Uganda and among young gay men in America fear of vaccine side effects was the main reason for not willing to participate [25, 26] while in America, ethnic minority communities reported fear that the trial vaccine would cause HIV infection [27]. Although HIV/AIDS awareness in the general population is high in Kenya, key population groups still face HIV stigma, discrimination

**Table 2. Predictors of willingness to participate in future HIV vaccine trial among the 214 participants in a simulated vaccine trial.**

| Predictor | Unadjusted | | | Adjusted | | |
|---|---|---|---|---|---|---|
| | OR | 95% CI | P-Value | aOR | 95% CI | P-Value |
| **Participant** | | | | | | |
| MSM | Ref | | | Ref | | |
| FSW | 0.93 | (0.41,2.13) | 0.86 | 1.00 | (0.38,2.67) | 0.996 |
| **Age, years** | | | | | | |
| 18–24 | Ref | | | Ref | | |
| 25–34 | 1.18 | (0.52, 2.67) | 0.69 | 1.45 | (0.59, 3.57) | 0.415 |
| ≥35 | 0.90 | (0.39, 2.11) | 0.81 | 1.16 | (0.42, 3.24) | 0.767 |
| **Highest level of education completed** | | | | | | |
| Primary | Ref | | | | | |
| Secondary | 1.48 | (0.61, 3.61) | 0.39 | | | |
| Tertiary/Higher | 1.11 | (0.42 3.00) | 0.83 | | | |
| **Alcohol use in the last month** | | | | | | |
| None | Ref | | | | | |
| Sometimes | 0.53 | (0.24, 1.16) | 0.11 | | | |
| **Alcohol use before sex in the last month** | | | | | | |
| Never | Ref | | | Ref | | |
| Sometimes | 0.46 | (0.23,0.92) | 0.03 | 0.55 | (0.25,1.24) | 0.148 |
| **Illicit drug use in the last month** | | | | | | |
| No | Ref | | | | | |
| Yes | 1.20 | (0.53, 2.71) | 0.66 | | | |
| **Alcohol CAGE score** | | | | | | |
| ≤1 | Ref | | | | | |
| ≥2 | 0.54 | (0.21, 1.41) | 0.21 | | | |
| **Number of sex partners last month** | | | | | | |
| ≤1 | Ref | | | | | |
| 2 | 0.67 | (0.25, 1.80) | 0.42 | | | |
| ≥3 | 0.75 | (0.32, 1.75) | 0.50 | | | |
| **Number of new sex partners in the last month** | | | | | | |
| 0 | Ref | | | | | |
| 1 | 0.95 | (0.35, 2.63) | 0.93 | | | |
| ≥2 | 0.74 | (0.35, 1.57) | 0.44 | | | |
| **Receptive anal Sex in the last month** | | | | | | |
| No | Ref | | | | | |
| Yes | 1.88 | (0.86, 4.09) | 0.11 | | | |
| **Insertive anal Sex in the last month** | | | | | | |
| No | Ref | | | | | |
| Yes | 0.43 | (0.14, 1.32) | 0.14 | | | |
| **Condom use in the last month** | | | | | | |
| Sometimes | Ref | | | Ref | | |
| Always | 1.68 | (0.76, 3.76) | 0.198 | 2.53 | (0.99, 6.45) | 0.050 |
| **Number of known HIV Behavioral Prevention methods** | | | | | | |
| 1 | Ref | | | | | |
| 2–3 | 0.75 | (0.26,2.17) | 0.601 | | | |
| 4–6 | 0.58 | (0.19,1.78) | 0.345 | | | |
| **Number of known HIV Biomedical Prevention methods** | | | | | | |
| 0 | Ref | | | | | |

*(Continued)*

**Table 2.** (Continued)

| Predictor | Unadjusted | | | | Adjusted | | |
|---|---|---|---|---|---|---|---|
| | OR | 95% CI | P-Value | | aOR | 95% CI | P-Value |
| 1–2 | 0.51 | (0.22,1.17) | 0.107 | | 0.38 | (0.15,0.93) | 0.035 |
| 3–4 | 0.44 | (0.17,1.15) | 0.095 | | 0.30 | (0.09,0.93) | 0.036 |
| Experience in SiVET study | | | | | | | |
| Good | Ref | | | | Ref | | |
| Very good | 3.91 | (1.86, 8.22) | <0.001 | | 5.54 | (2.39, 12.89) | <0.001 |

and violence, all of which may deter people from participation because it is related to HIV/ AIDS [4, 9, 12].

Our findings are comparable to those from a study in China among MSM where 77% reported willingness [28]. Although these participants had not just undertaken a simulated trial, this study used computer assisted self-interview method to assess willingness to participate, which may have reduced response bias (i.e., the participants may have been less inclined to say what they thought the investigators "wanted to hear"). However, our level of willingness was lower than findings in other studies, i.e.91% of willingness among a cohort of young gay men in America [26], 94% willingness among healthy low risk adults in an observational cohort in Kenya [15], 95% in a rural community-based cohort study in Uganda [14] and 99% among high risk men and women from fishing communities cohort study in Uganda [19]. This difference in willingness rates could be due to differences in study population, and/or study procedures. Unlike the five other studies [14, 15, 19, 26, 28], our study population had just participated in a simulated clinical trial, thus our study's participants' expression of willingness was less hypothetical hence may have been a more accurate reflection of actual willingness to participate.

Although participants' willingness to participate is essential for the enrolment in trials, high levels of reported willingness to participate may not necessarily translate into actual trial participation. A Kenyan study that compared hypothetical and actual willingness to participate in the same clinic as our study found that only 30% of the participants who had previously expressed hypothetical willingness to participate in HIV vaccine trials actually presented for screening in the subsequent vaccine trial [15]. These were volunteers that had participated in a cohort study, but not one designed to mimic the rigors of clinical trial participation (they had fewer follow up visits and no vaccines were administered).

Our study had some limitations. The sample size of 46 FSW was small and does not allow for statistical comparison with MSM, and also does not allow many findings or conclusions to be made about FSW participation and willingness. In this study we were not required to collect data regarding MSM sex work hence not able to assess the willingness to participate of those

**Table 3. Motivating factors for future participation in HIV Vaccine trials among 174 SiVET participants reporting willingness to participate.**

| Motivator | No. Responses | % |
|---|---|---|
| Education about HIV | 103 | 59 |
| Healthcare | 99 | 57 |
| Altruism | 69 | 40 |
| Regular HIV VCT | 69 | 40 |
| Hope of being protected against getting HIV | 64 | 37 |
| Other responses including Reimbursement, liking the staff, an activity to keep them busy | 34 | 20 |

engaged in sex worker. The Hepatitis B vaccine used in the SiVET is already licenced hence it might not fully represent the experience involving a trial with an experimental HIV vaccine; the experience however, realistically unpackages and demystifies actual clinical trial participation, hence may serve as a more valuable gauge of willingness compared to hypothetical scenarios where participants are simply told about clinical trials.

## Conclusions

The study demonstrated a high level of willingness to participate in future HIV vaccine trials among MSM and FSW in Nairobi, Kenya, after participating in a study designed to mimic the rigors of a clinical trial. This suggests that these groups might be a reliable target population for inclusion in future HIV vaccine trials. Assessment of willingness to participate in these populations provides important information that may help inform future education and recruitment efforts for vaccine trials. Improving the research experience for members of key populations could help impact their willingness to participate in HIV vaccine trials.

## Supporting information

**S1 File. PDF Risk assessment questionnare.**
(PDF)

**S2 File. PDF Vaccine Trial Willingness questionnare.**
(PDF)

## Acknowledgments

We wish to acknowledge administrative support from University of California, San Francisco's International Traineeships in AIDS Prevention Studies (ITAPS). As part of the ITAPS we thank the helpful input in the manuscript formulation of Willi McFarland, Claudia Di Lorenzo, Lucy Chimonyi and Dr Tanesha Hickmann. We also appreciate the assistance of Robert Langat of KAVI-ICR. We would like to acknowledge the trial participants, the investigators, KAVI-ICR, SWOP-Kenya and IAVI clinical development teams.

## Author Contributions

**Conceptualization:** Elizabeth Mueni Mutisya.

**Data curation:** Elizabeth Mueni Mutisya, Delvin Nyasani, Hannah Nduta, Andrew Abaasa.

**Formal analysis:** Elizabeth Mueni Mutisya, Andrew Abaasa.

**Funding acquisition:** Aggrey Omu Anzala.

**Investigation:** Joshua Kimani, Aggrey Omu Anzala.

**Methodology:** Elizabeth Mueni Mutisya.

**Project administration:** Joshua Kimani, Aggrey Omu Anzala.

**Resources:** Aggrey Omu Anzala.

**Supervision:** Vincent Muturi-Kioi, Krysia Lindan, Aggrey Omu Anzala.

**Validation:** Vincent Muturi-Kioi, Krysia Lindan.

**Writing – original draft:** Elizabeth Mueni Mutisya.

**Writing – review & editing:** Elizabeth Mueni Mutisya, Gaudensia Mutua, Delvin Nyasani, Hannah Nduta, Rhoda W. Kabuti, Vincent Muturi-Kioi, Gloria Omosa-Manyonyi, Andrew Abaasa, Krysia Lindan, Matt A. Price, Joshua Kimani, Aggrey Omu Anzala.

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
