## [Decision Letter · Decision Letter 0]

6 Jan 2020

PONE-D-19-31011

Willingness to participate in future HIV vaccine trials among men who have sex with men and female sex workers living in Nairobi, Kenya.

PLOS ONE

Dear Ms Mutisya,

Thank you for submitting your manuscript to PLOS ONE. After careful consideration, we feel that it has merit but does not fully meet PLOS ONE’s publication criteria as it currently stands. Therefore, we invite you to submit a revised version of the manuscript that addresses the points raised during the review process.

The reviewers both raise valid points that need to be addressed. In particular, Reviewer 2 indicates the need for clarifications regarding the Methods, including sampling, participants, data collection/questionnaire, and data analysis. Both reviewers make helpful suggestions about the introduction and discussion sections, including the need for additional reporting of your own findings and then for discussing these in the context of other related research, as well as suggested revision to the conclusion.

We would appreciate receiving your revised manuscript by Feb 20 2020 11:59PM. To enhance the reproducibility of your results, we recommend that if applicable you deposit your laboratory protocols in protocols.io, where a protocol can be assigned its own identifier (DOI) such that it can be cited independently in the future. For instructions see: http://journals.plos.org/plosone/s/submission-guidelines#loc-laboratory-protocols

We look forward to receiving your revised manuscript.

Kind regards,

Peter A Newman, Ph.D

Academic Editor

PLOS ONE

Reviewers' comments:

Reviewer's Responses to Questions

**Comments to the Author**

1. Is the manuscript technically sound, and do the data support the conclusions?

Reviewer #1: Partly

Reviewer #2: Yes

2. Has the statistical analysis been performed appropriately and rigorously? 

Reviewer #1: I Don't Know

Reviewer #2: Yes

3. Have the authors made all data underlying the findings in their manuscript fully available?

Reviewer #1: Yes

Reviewer #2: Yes

4. Is the manuscript presented in an intelligible fashion and written in standard English?

Reviewer #1: Yes

Reviewer #2: Yes

5. Review Comments to the Author

Reviewer #1: Reviewer’s comments

The presented data is timely in the efforts to recruit key populations in HIV vaccine trials. Targeting men who have sex with men and female sex workers for future trials is essential because these are the most affected groups in SSA. Nevertheless, some parts of the manuscript need some revisions to increase clarity. These are:

Abstract

The conclusion lines 44-47 needs revision. This statement does not match the key results. The authors should modify this statement to increase clarity.

Introduction

The first paragraph, from line 54… should start with the current global, regional, and national HIV prevalence, followed by specific key populations' HIV prevalence. Part of this description appears between lines 60 and 66.

Discussion

The authors should discuss their results before relating to others' work. See, for example, lines 215 – 216; what does the willingness of four of five respondents mean in the context of this study?

Line 257-261, access to health services as a motivator to take part in HIV vaccine trials have been reported from other low-income countries besides Uganda and Kenya. It would be nice to expand this particular discussion by using additional related citations.

Additionally, the authors must discuss the limitations and mitigations of this study. For example, the questionnaire was administered 12-15 months after the trial. Would this gap in terms of timing influence the obtained results?

Conclusion

Lines 275 – 281, should be rephrased to match the current key study results. For example, the issue of improving recruitment and retention was not part of the key results. Although it is an important recommendation, I think other issues about the current results should be emphasized first.

Reviewer #2: The manuscript focuses on assessing the willingness to participate in future vaccine trials among MSM and female sex workers in Kenya. As the study assessed the willingness among the participants of a SiVET, it is claimed that the reported willingness maybe closer to the willingness to participate in actual trials.

Major comments:

1. The first sentence in the Introduction (line numbers 54, 55) seems to make a sweeping statement about achievement of success in care, treatment and prevention. This statement needs to be qualified and specified – i.e., which achievements are being referred to – biomedical or public health achievements?

2. It is not clear whether and what percentage of MSM participants engaged in sex work. That information may be important for future trials.

3. Given the findings that those with (good) knowledge about HIV biomedical prevention methods had lower odds to participate in future trials, it is important to know what were these prevention methods that were asked in the questionnaire (e.g., PrEP, microbicides?) Whether this information (on prevention methods) was provided as part of SiVET?

4. Sample size calculation: How the decision to enroll 200 MSM and 50 FSW was arrived at from the calculated sample size of 250?

5. It is not clear what is meant by “…included in a backward elimination algorithm of multivariable logistic regression to arrive at a final model, retaining those variables with a p <0.05”. Whether it means only those variables with p <0.05 were included in the final multivariable model? But it does not seem to be the case (Table 2).

6. Whether the five new infections occurred among MSM or FSWs?

7. In the discussion section, the authors could comment on the reported reasons for not willing to participate: “time commitment, fear that their HIV status would be disclosed to others without their permission and the strict contraception requirements”. For example, the prevalence of HIV-related stigma/discrimination in Kenya seems to be associated with the second concern.

Minor comment:

There are minor language errors (e.g., line 146: “participants’ were classified”, which can be corrected by careful proofreading.

6. PLOS authors have the option to publish the peer review history of their article (what does this mean?). If published, this will include your full peer review and any attached files.

Reviewer #1: No

Reviewer #2: No

---

## [Author Response · Author response to Decision Letter 0]

26 May 2020

19-Feb-2020

Re: PONE-D-19-31011

Willingness to participate in future HIV vaccine trials among men who have sex with men and female sex workers living in Nairobi, Kenya.

Dear Dr. Newman, 

Thank you for your comments on our manuscript. Below, we list each comment, with our responses shown. We look forward to hearing from you soon.

Sincerely,

Elizabeth Mutisya

 Authors’ response: We have reviewed the above and confirm we meet these requirements

 Authors’ response: Thank you for this comment; we have now included a copy of our questionnaire uploaded as supporting information. (S1 File PDF Risk assessment questionnaire)

Authors’ response: The KAVI-ICR agrees to open data access. The data collected by KAVI-ICR can be made available to other bona fide researchers, upon application to access the necessary data from which this manuscript was generated. The corresponding and other co-author emails have been provided and they could be contacted anytime for further clarifications and/or support to access the data.

Reviewer #1: Reviewer’s comments

The presented data is timely in the efforts to recruit key populations in HIV vaccine trials. Targeting men who have sex with men and female sex workers for future trials is essential because these are the most affected groups in SSA. Nevertheless, some parts of the manuscript need some revisions to increase clarity. These are:

Abstract

The conclusion lines 44-47 needs revision. This statement does not match the key results. The authors should modify this statement to increase clarity.

Authors’ response: We have changed the language some to better reflect our findings in the results.

Introduction

The first paragraph, from line 54… should start with the current global, regional, and national HIV prevalence, followed by specific key populations' HIV prevalence. Part of this description appears between lines 60 and 66.

Authors’ response: Thank you for your comment. While we appreciate that it often seems like most HIV clinical research papers start this way, the important take home message here is that we are not on target to achieve upcoming milestones to reduce incidence, and that an HIV vaccine remains an important goal. To do this, we need volunteers for upcoming efficacy trials. We feel our research supports the inclusion of these key populations in future trials.

Discussion

The authors should discuss their results before relating to others' work. See, for example, lines 215 – 216; what does the willingness of four of five respondents mean in the context of this study?

Authors’ response: We have added another sentence to the beginning of the discussion text prior to discussing others’ work.

Line 257-261, access to health services as a motivator to take part in HIV vaccine trials have been reported from other low-income countries besides Uganda and Kenya. It would be nice to expand this particular discussion by using additional related citations.

additionally, the authors must discuss the limitations and mitigations of this study. For example, the questionnaire was administered 12-15 months after the trial. Would this gap in terms of timing influence the obtained results?

Authors’ response: Thank you for these comments. We have included references outside of Uganda and Kenya. We also kindly refer the reviewer to the last paragraph of the discussion, where we summarize limitations and mitigations. We also respectfully remind the reviewer that the questionnaire was administered 12-15 months after enrollment, on the last study visit. 

Conclusion

Lines 275 – 281, should be rephrased to match the current key study results. For example, the issue of improving recruitment and retention was not part of the key results. Although it is an important recommendation, I think other issues about the current results should be emphasized first.

Authors’ response: Thank you for these comments. In this paragraph, we aren’t talking about improving recruitment and retention, instead we are noting that those who reported a better experience in the SiVET were more likely to report being willing to participate, suggesting that participants who felt their needs met were probably more likely to enroll into a future trial. Perhaps I am not understanding your question? 

 Reviewer #2: The manuscript focuses on assessing the willingness to participate in future vaccine trials among MSM and female sex workers in Kenya. As the study assessed the willingness among the participants of a SiVET, it is claimed that the reported willingness maybe closer to the willingness to participate in actual trials.

Major comments:

1. The first sentence in the Introduction (line numbers 54, 55) seems to make a sweeping statement about achievement of success in care, treatment and prevention. This statement needs to be qualified and specified – i.e., which achievements are being referred to – biomedical or public health achievements?

Authors’ response:

We have stated in the introduction section that “there are various approaches available to public health including behavioural, structural and biomedical intervention” 

2. It is not clear whether and what percentage of MSM participants engaged in sex work. That information may be important for future trials.

Authors’ response: It would have been valuable information, however in this particular study we were not required to collect data regarding sex work.

3. Given the findings that those with (good) knowledge about HIV biomedical prevention methods had lower odds to participate in future trials, it is important to know what were these prevention methods that were asked in the questionnaire (e.g., PrEP, microbicides?) Whether this information (on prevention methods) was provided as part of SiVET?

Authors’ response: Both biomedical and behavioural HIV prevention methods were included in the questionnaire. We have also included the specific HIV preventive methods have also been in the text.

We have included the risk assessment questionnaire as an attachment. The information on prevention methods provided is found on page 3numbers 10 and 11. (S1 File PDF Risk assessment questionnaire)

4. Sample size calculation: How the decision to enroll 200 MSM and 50 FSW was arrived at from the calculated sample size of 250?

Authors’ response: Our decision to enroll 250 volunteers, 20% of whom would be FSW by design, was also based on feasibility considerations. We were most interested in reaching out to the MSM communities in Nairobi where HIV risk remains high, however our partner SWOP works with both male and female sex workers. To avoid stigma and bad feelings with our partners, we agreed to enroll a certain percentage of women into the study and settled on 20%. 

5. It is not clear what is meant by “…included in a backward elimination algorithm of multivariable logistic regression to arrive at a final model, retaining those variables with a p <0.05”. Whether it means only those variables with p <0.05 were included in the final multivariable model? But it does not seem to be the case (Table 2).

Authors’ response: The decision to keep a variable in the multivariable model was dependent on the model log likelihood ratio test p-value for the inclusion of a given variable and not Wald p-value, except for age and trial population as these were considered a priori confounders. 

We have corrected the text in our methods to make it clearer “Univariate logistic regression was used to evaluate the association of all variables with willingness to participate; factors associated with willingness to participate at p<0.20 on log likelihood test were included in the initial multivariable model (except for age and trial population that were included a priori). Factors were retained in the multivariable model if the log likelihood test p-value of inclusion of a factor was less than 0.05”.

6. Whether the five new infections occurred among MSM or FSWs?

Authors’ response: The five new HIV infections occurred among MSM.

We have added the information text in the results section and Fig 1 HIV infected. “5M: 0F”

7. In the discussion section, the authors could comment on the reported reasons for not willing to participate: “time commitment, fear that their HIV status would be disclosed to others without their permission and the strict contraception requirements”. For example, the prevalence of HIV-related stigma/discrimination in Kenya seems to be associated with the second concern.

Authors’ response: Information on the reported reasons for not willing to participate has been added in the discussion section. Additional related citations have been included in the reference.

We have expanded the discussion citing Newman PA, Duan N, Roberts KJ, Seiden D, Rudy ET, Swendeman D, et al. HIV vaccine trial participation among ethnic minority communities: barriers, motivators, and implications for recruitment. J Acquir Immune D

To comment on the “fear that their HIV status would be disclosed to others without their permission“ indicates stigma/discrimination and we have expanded in the discussion section that “although HIV/AIDS awareness is high in Kenya, key population groups still face HIV stigma, discrimination and violence hence, this may deter people from participation because it is related to HIV/AIDS“

Minor comment:

There are minor language errors (e.g., line 146: “participants’ were classified”, which can be corrected by careful proofreading.

 Authors’ response: We have reviewed the paper again, and there are minor corrections and clarifications throughout that we hope make it clearer.

---

## [Editor Report · Decision Letter 1]

22 Jun 2020

PONE-D-19-31011R1

Willingness to participate in future HIV vaccine trials among men who have sex with men and female sex workers living in Nairobi, Kenya.

PLOS ONE

Dear Dr. Mutisya,

Thank you for submitting your manuscript to PLOS ONE. After careful consideration, we feel that it has merit but does not fully meet PLOS ONE’s publication criteria as it currently stands. Therefore, we invite you to submit a revised version of the manuscript that addresses the points raised during the review process.

The authors have made a few of the revisions suggested by the two reviewers, and these are helpful. However, several of the reviewers' comments are not adequately responded to; and some are only responded to in the Reviewer's comments, with no corresponding changes in the manuscript. This is not sufficient. Also, please indicate the exact Line number in the manuscript where each revision has been made to ensure the reviewer/editor can locate them. This is more important since a few of the comments seem not to be responded to.

For example, Reviewer 1: "The first paragraph...should start with current global, regional and national prevalence." The response to the reviewer is inadequate.

In the conclusion: Your response, "Perhaps I am not understanding your question" is unfortunately not helpful. Both reviewers have suggested that you stay closer to your findings when you make recommendations based on your study.

Reviewer 2:

1. The added sentence is too general and thus does not adequately address the reviewer's comment.

2. The response is ok, but you must now add this as a specific study limitation.

3. "We have also included the specific HIV preventive methods in the text."  Please specifically indicate where these are these in the text.

4. Please indicate that you have incorporated this explanation into the manuscript text (not only in response to the reviewer) and indicate the line number where it appears in the revised manuscript.

7. The reference mentioned as "expanded the discussion citing" is in fact nowhere in the reference list. Also, simply adding 1 reference does not adequately address the reviewer's comment.

Finally, you have included several comments that you added to your track changed version, however you have not in fact made these changes; it appears that a number of newly added references are missing. Please carefully revise and proofread before resubmission, and make sure to actually add the new references as citations in the text, and to the reference list!

Line 277 "insert to include refs 11& 12" ??

Line 320 - 321  You left in (REF) and (REF). You must actually include the reference number here; and the newly added references must be indicated in the Reference list at the end. There are no apparent changes however to the reference list, despite the claim that a few new references have been added. 

Line 328:"insert ref number 3" ??

We look forward to receiving your revised manuscript.

Kind regards,

Peter A Newman, Ph.D

Academic Editor

PLOS ONE

---

## [Author Response · Author response to Decision Letter 1]

31 Jul 2020

30th Jul 2020

RE: PONE-D-19-31011R1

Willingness to participate in future HIV vaccine trials among men who have sex with men and female sex workers living in Nairobi, Kenya.

Dear Dr. Newman, 

Thank you for your comments on our manuscript. Below, we list each comment with our responses shown. We look forward to hearing from you soon.

Sincerely,

Elizabeth Mutisya

The authors have made a few of the revisions suggested by the two reviewers, and these are helpful. However, several of the reviewers' comments are not adequately responded to; and some are only responded to in the Reviewer's comments, with no corresponding changes in the manuscript. This is not sufficient. Also, please indicate the exact Line number in the manuscript where each revision has been made to ensure the reviewer/editor can locate them. This is more important since a few of the comments seem not to be responded to.

Authors’ response: Thank you for the helpful suggestions. We have reviewed the manuscript and confirm corresponding changes to the document. We have indicated the exact line number where revision has been made.

For example, Reviewer 1: "The first paragraph...should start with current global, regional and national prevalence." The response to the reviewer is inadequate.

Authors’ response: Thank you for the clarification. We have included the current global, regional and national HIV prevalence and specific key population HIV prevalence in the first paragraph of the introduction and expanded the citation. Lines 51-70

Abstract

The conclusion lines 44-47 needs revision. This statement does not match the key results. The authors should modify this statement to increase clarity 43-46

Authors’ response: Thank you for the comment. We have changed the language to better emphasize our study findings lines 44-47

In the conclusion: Your response, "Perhaps I am not understanding your question" is unfortunately not helpful. Both reviewers have suggested that you stay closer to your findings when you make recommendations based on your study.

Authors’ response: Thank you for the clarification. We have changed the language to better emphasize our study findings. Lines 316-319

Discussion

The authors should discuss their results before relating to others' work. See, for example, lines 215 – 216; what does the willingness of four of five respondents mean in the context of this study?

Authors’ response: We have rearranged the paragraphs in the discussion section to first discuss the results of this study before relating to others’ work, from lines 235-275

Reviewer 2:

1. The added sentence is too general and thus does not adequately address the reviewer's comment.

Authors’ response: Thank you for the clarification. We have added the global, regional and national HIV prevalence and specific key population HIV prevalence in the first paragraph of the introduction to qualify the HIV successes and challenges. Lines 51-70

2. The response is ok, but you must now add this as a specific study limitation.

Authors’ response: Thank you for the reminder. We have included information not collected regarding MSM sex work as a study limitation. Lines 304-306

3. "We have also included the specific HIV preventive methods in the text." Please specifically indicate where these are these in the text.

Authors’ response: This information did not appear in the initial manuscript. However, in the revised manuscript we had included the specific biomedical and behavioral HIV preventive methods in lines 142-147

4. Please indicate that you have incorporated this explanation into the manuscript text (not only in response to the reviewer) and indicate the line number where it appears in the revised manuscript.

Authors’ response: Please find the following sample size calculation information under METHODS -SAMPLE SIZE lines 103-107

A sample size of 250 for the SiVET was calculated based on being able to estimate a one-year retention rate of 80% with a precision of ±5%. The target was to enrol 200 MSM and 50 FSW. The choice to enrol 20% FSW was deliberate because HIV incidence is higher among Kenyan MSM than Kenyan FSW; we also wished to avoid stigma and any hard feelings among our partners at SWOP-Kenya and agreed that enrolling some women would be appropriate. 

7. The reference mentioned as "expanded the discussion citing" is in fact nowhere in the reference list. Also, simply adding 1 reference does not adequately address the reviewer's comment.

Authors’ response: Thank you for the comment. Reported reasons for not willing to participate have been added in the discussion section of the revised manuscript and the related citations have been included in the reference list. For ease of access please find this information in lines (273-280) and reference list numbers (25, 26, 27, 4, 9 & 12)

Finally, you have included several comments that you added to your track changed version, however you have not in fact made these changes; it appears that a number of newly added references are missing. Please carefully revise and proofread before resubmission, and make sure to actually add the new references as citations in the text, and to the reference list!

Authors’ response: Thank you for the comments regarding track changed version. We have reviewed the paper again, and there are corrections and clarifications throughout that we hope make it clearer. The reference list has been reviewed and can confirm the additional citations have been included.

Line 277 "insert to include refs 11& 12" ??

Authors’ response: These added citations appear in lines (288& 291) of the revised manuscript and are included in the reference list numbers 14 &15.

Line 320 - 321 You left in (REF) and (REF). You must actually include the reference number here; and the newly added references must be indicated in the Reference list at the end. There are no apparent changes however to the reference list, despite the claim that a few new references have been added. 

Authors’ response: The lines mentioned 320-321 appear in a paragraph with information related to reported motivators for participation now lines 263-272 in the revised manuscript. The citations reflecting this information is references numbers 19,22,23,16 &, 24.

Line 328:"insert ref number 3" ??

Authors’ response: The line 328 appears in a paragraph with information related to reasons for not willing to participate in future trial. This information now appears in lines 273-280 and the references to the information are numbers 4,9,12, 25, 26 & 27

Authors’ response: We have reviewed the above and included the above items in our submission.

Authors’ response: We have reviewed and made changes in the financial disclosure. The updated statement is included in the cover letter. 

We have reviewed the guidelines of the figure file and confirm we meet these requirements.

Authors’ response: We did review the above and confirm we meet these requirements. Our URL https://doi.org/10.3886/E119622V1

 We have also reviewed and expanded the acknowledgment section to include persons/teams who offered their assistance lines 321-325

We have also reviewed and included Supporting information S1 Fig.tif Fig 1 in lines 414

---

## [Editor Report · Decision Letter 2]

10 Aug 2020

Willingness to participate in future HIV vaccine trials among men who have sex with men and female sex workers living in Nairobi, Kenya.

PONE-D-19-31011R2

Dear Dr. Mutisya,

We’re pleased to inform you that your manuscript has now been judged scientifically suitable for publication and will be formally accepted for publication once it meets all outstanding technical requirements.

Kind regards,

Peter A Newman, Ph.D

Academic Editor

PLOS ONE
---

## [Editor Report · Acceptance letter]

13 Aug 2020

PONE-D-19-31011R2 

Willingness to participate in future HIV vaccine trials among men who have sex with men and female sex workers living in Nairobi, Kenya. 

Dear Dr. Mutisya:

I'm pleased to inform you that your manuscript has been deemed suitable for publication in PLOS ONE. Congratulations! Your manuscript is now with our production department. 

Kind regards, 

on behalf of

Dr. Peter A Newman 

Academic Editor

PLOS ONE